# Acceleration of an Electron Bunch with a Non–Gaussian Transverse Profile in Proton-Driven Plasma Wakefield

**Linbo Liang** [1,2,*] , **Guoxing Xia** [1,2,*] , **Alexander Pukhov** [3] **and John Patrick Farmer** [4,5]

1 Department of Physics and Astronomy, University of Manchester, Manchester M13 9PL, UK
2 Cockcroft Institute, Warrington, WA4 4AD, UK
3 Institut für Theoretische Physik I, Heinrich-Heine-Universität, 40225 Düsseldorf, Germany
4 CERN, 1211 Geneva, Switzerland
5 Max Planck Institute for Physics, 80805 Munich, Germany
* Correspondence: linbo.liang@postgrad.manchester.ac.uk (L.L.); guoxing.xia@manchester.ac.uk (G.X.)

**Abstract:** Beam-driven plasma wakefield accelerators typically use the external injection to ensure controllable beam quality at injection. However, the externally injected witness bunch may exhibit a non-Gaussian transverse density distribution. Using particle-in-cell simulations, we show that the common beam quality factors, such as the normalized RMS emittance and beam radius, do not strongly depend on the initial transverse shapes of the witness beam. Nonetheless, a beam with a highly-peaked transverse spatial profile can achieve a higher fraction of the total beam charge in the core. The same effect can be seen when the witness beam's transverse momentum profile has a peaked non-Gaussian distribution. In addition, we find that an initially non-axisymmetric beam becomes symmetric due to the interaction with the plasma wakefield.

**Keywords:** plasma wakefield; PWFA; non-Gaussian; transverse dynamics; particle-in-cell simulation

## 1. Introduction

Beam-driven plasma wakefield accelerators (PWFAs) have shown the ability to generate ultra-high accelerating gradients ($\sim$GV/m), which far exceeds those in radio-frequency based accelerators [1,2]. Among the currently available beam drivers, proton beams from the CERN accelerator complex such as the Large Hadron Collider (LHC) or Super Proton Synchrotron (SPS) stand out to be the most promising driver for TeV-level electron acceleration in a single plasma stage [3]. The Advanced Wakefield Experiment (AWAKE) at CERN is a proof-of-principle proton beam driven plasma wakefield experiment [4]. In AWAKE, the long SPS proton bunch, with a typical RMS length $\sigma_z$ of 6–12 cm and energy of 400 GeV, is first divided into a series of micro-bunches under the effect of seeded self-modulation [5,6], which then effectively drive $\sim$GV/m-level plasma wakefields.

The AWAKE Run 1 experiment has demonstrated the acceleration of externally injected, 18 MeV electrons to the energy of 2 GeV in 2018 [4]. To achieve a better control of the electron beam quality during acceleration, the AWAKE Run 2 (2021-) plans to use a separate plasma stage for the electron acceleration after the proton self-modulation stage [7,8]. In the acceleration stage, an electron bunch is injected as the witness to load on the quasi-linear wakefield driven by the self-modulated proton bunch train. The aim of beam loading is to flatten the longitudinal wakefield along the witness beam so as to reduce the energy spread during the acceleration. To achieve this goal, one need to choose appropriate witness beam parameters, including the bunch charge, length and injection (or loading) position [9]. In addition, the concepts of beam matching will also be implemented in the Run 2 electron acceleration [7,8]. The idea of beam matching is to match the beam divergence force with the plasma focusing force to prevent the emittance growth due to the collective beam electron oscillations. For a dense enough witness beam, it is able to fully expel the plasma electrons from the beam propagation axis and form an electron-free bubble area [10]. The

radially linear focusing force inside the plasma bubble, which is proportional to the radial offset $r$, i.e., $F_\perp \propto r$, can preserve the slice emittance of the electron beam. If the RMS beam radius of the witness beam satisfies the matching condition, the projected beam emittance growth can be also suppressed [7].

In particle accelerators, beam profiles are typically assumed to be Gaussian. However, due to the action of non-linear forces such as the space charge effect or various nonlinearities in the beam line, realistic particle beams normally deviate from the standard Gaussian shape [11–14]. A non-Gaussian transverse beam distribution can affect the beam properties, such as the transverse emittance and brightness [15]. Here we look into this intrinsic mechanism and explore the influences of non-Gaussian beam profiles on electron beam dynamics in a quasilinear wakefield. This work will be crucial for the optimisation of accelerators and for the development of diagnostics for beams with more realistic distributions.

This paper is organized as follows. The simulation configuration and main parameters are presented in Section 2. In Section 3, the plasma wakefield properties and the basic beam dynamics of a Gaussian profile witness bunch are shown. The mathematical characterizations of the higher-order features of non-Gaussian distributions as well as their influences on the plasma response and beam properties, e.g., emittance and brightness, are discussed in Section 4. Finally, we summarize all key findings in Section 5.

## 2. Simulation Configurations

The scope of this work is to focus on the witness beam dynamics in a stable wakefield. It's therefore convenient to use the toy model that was first introduced by Olsen et al. [7]. It employs a single, non-evolving "proton" bunch rather than the self-modulated proton bunch train as the driver travelling in an initially homogeneous plasma and an externally injected electron bunch as the witness beam trailing behind, as shown in Figure 1. This model can significantly reduce the simulation cost since we do not need to simulate the proton self-modulation every time, while the latter can be ultra time-consuming.

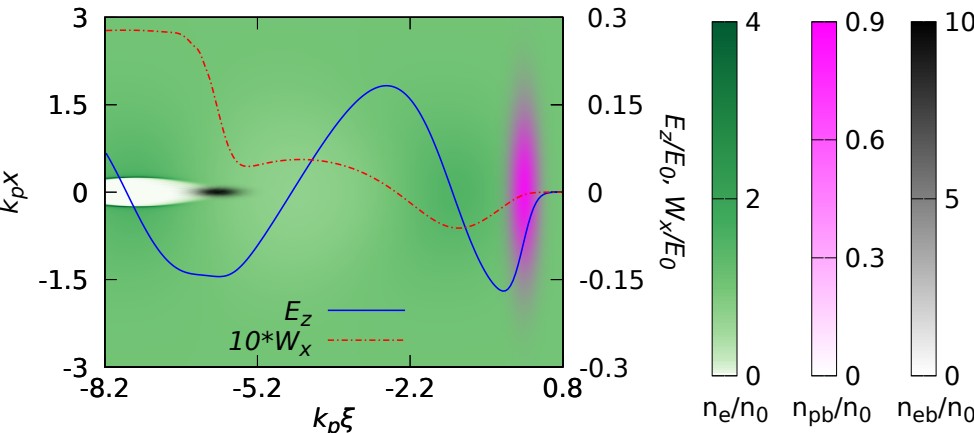

**Figure 1.** The simplified scheme of AWAKE Run 2 acceleration stage. The densities of plasma electrons ($n_e$, green), the proton driver bunch ($n_{pb}$, pink) and the electron witness bunch ($n_{eb}$, black) are shown in colourmaps, with values normalized by the unperturbed plasma electron density $n_0 = 7 \times 10^{14}$ cm$^{-3}$. Particle beams propagate from left to right. $\xi = z - ct$ is the longitudinal coordinate in the co-moving frame. The blue solid line represents the loaded longitudinal wakefield $E_z$, and the red dashed line is the transverse wakefield $W_x = E_x - cB_y$ at one $\sigma_{re}$ off the longitudinal axis. $\sigma_{re}$ is the RMS beam radius of the electron witness bunch.

The nominal density of the uniform plasma is $n_0 = 7 \times 10^{14}$ cm$^{-3}$. The non-evolving proton driver has a Lorentz factor of $\gamma_{p_0} = 426.29$, an RMS bunch length of $\sigma_{zp} = 40\,\mu$m, an RMS transverse size of $\sigma_{rp} = 200\,\mu$m and a charge of 2.34 nC. The driver parameters are the same as those in Ref. [7], which allow us to mimic the quasi-linear wakefield driven by the self-modulated SPS proton bunches.

The baseline witness electron beam in this study has the following parameters: a charge of $Q_e = 120$ pC, an RMS bunch length of $\sigma_{ze} = 60$ µm, an initial energy of 150 MeV ($\gamma_{e_0} = 295.54$) and an initial normalized emittance of $\epsilon_{n_0} = 6.84$ µm. The change of baseline witness beam parameters compared with those in previous studies [7,8] is a result of the evolution of the electron beamline design [14,16]. The increase in the initial emittance is due to the Coulomb scattering of the witness electrons when they penetrate through two aluminium foils before the injection point (one for the vacuum window and the other one for laser beam dump) [17]. The witness bunch radius at the injection point is chosen as the matched radius in the pure plasma ion column, which is given by [18]:

$$\sigma_{r,ic} = \left( \frac{2\epsilon_{n0}^2}{\gamma_{e0}k_p^2} \right)^{1/4}, \tag{1}$$

where $\epsilon_{n0}$ is the normalized emittance at the injection point, and $\gamma_{e0}$ is the Lorentz gamma factor of the electron beam. The plasma wave number $k_p$ is given as $k_p = \omega_p/c$, where $c$ is the speed of light. $\omega_p = \sqrt{n_0 e^2/m_0 \varepsilon_0}$ is the plasma frequency, $\varepsilon_0$ is the vacuum permittivity, $m_0$ and $e$ are the rest mass and charge of an electron, respectively. For the witness parameters we considered here, Equation (1) gives the same matched beam size $\sigma_{r,ic} = 10.64$ µm for both the low-charge (120-pC) and high-charge (400-pC) cases. Then the peak density $n_{e0} = N_e/((2\pi)^{3/2}\sigma_{ze}\sigma_{r,ic}^2)$ of a Gaussian bunch for the two cases can be calculated as 10 (120 pC) and 20 (400 pC) times the plasma density $n_0$, respectively. Here, $N_e = Q_e/e$ is the number of bunch electrons. Alternatively, for the bubble-regime, a more significant parameter for evaluating the bubble formation could be the normalized beam charge $\tilde{Q} = N_e k_p^3/n_0$ [19]. Then for the 120-pC and 400-pC electron beam, the normalized charges are calculated as 0.13 and 0.44, respectively. For both cases, the witness charge is large enough to drive a plasma bubble, but the bubble formation may not complete, as shown in Figure 1. The default delay between the two bunches is set as $k_p \Delta \xi = k_p (\xi_{0p} - \xi_{0e}) = 6$ at the beginning of the simulation, where $\xi_{0p}$ and $\xi_{0e}$ are the initial longitudinal centroids of the proton and electron bunches in the co-moving frame, respectively.

Numerical simulations in this paper are mainly carried out with the two-dimensional (2D) axisymmetric quasi-static particle-in-cell (PIC) code LCODE [20]. The simulation window co-moving with the particle bunches (with the speed of light $\sim c$) has the similar dimensions as shown in Figure 1, and it is represented in the 2D cylindrical geometry $(z, r)$. The cell size is $(0.01 \times 0.01)k_p^{-1}$ in both the $z-$ and $r-$direction. The time step is $\omega_p^{-1}$, which is enough to resolve the betatron motion of witness electrons. The witness beam is simulated with $10^6$ equally-weighted macro-particles.

## 3. Beam Dynamics of a Gaussian Electron Bunch

Figure 2 shows the 2D colourmaps ($y = 0$) of the plasma wakefields for the case with a low-charge (120 pC) witness beam. The initial charge distribution of the witness beam is Gaussian in both the transverse and longitudinal directions. It can be seen that the longitudinal wakefield $E_z$ is nearly constant in the vicinity of the witness beam as shown in Figure 2a. Its average value also remains almost unchanged due to a very small amount of dephasing over the 10-m propagation distance. Therefore, the average beam energy increases linearly. However, since the accelerating gradient is not fully constant along the whole witness beam, it leads to a finite energy spread after acceleration, as shown in Figure 3a.

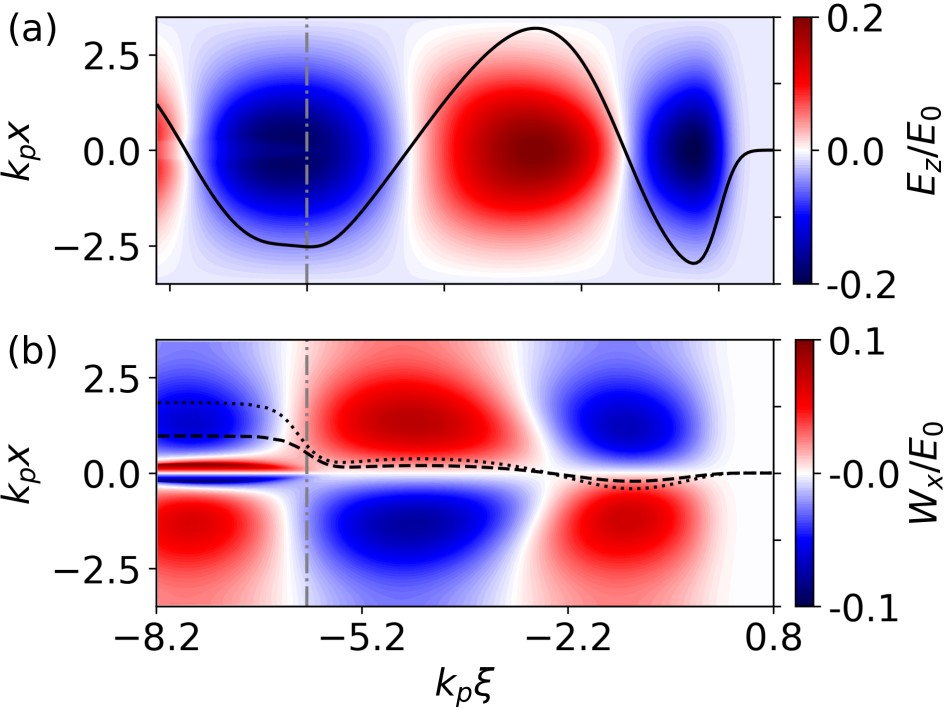

**Figure 2.** Plasma wakefields for the low-charge (120 pC) case in colourmaps. (**a**) The longitudinal wakefield $E_z$. The 1D lineout shows the on-axis value of $E_z$, whose value range is the same as the right colorbar. (**b**) The transverse wakefield $W_x = E_x - cB_y$. The dashed line and the dotted line show the value at $r = \sigma_{re}$ and $r = 2\sigma_{re}$, respectively. The initial loading position $\xi_{0e}$ (or the longitudinal centroid) of the witness electron bunch, shown by vertical grey dash-dotted lines in both plots.

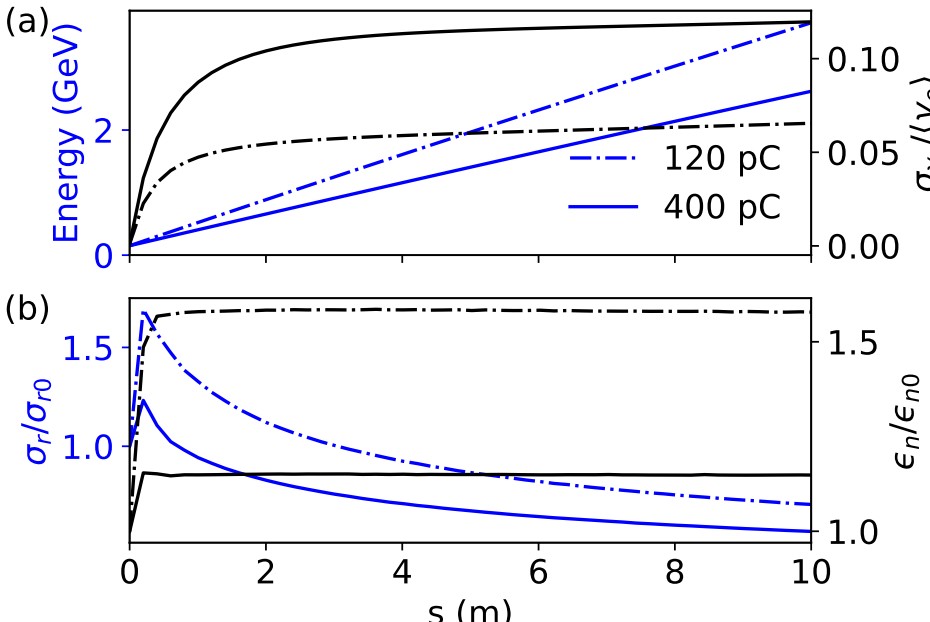

**Figure 3.** Evolution of the Gaussian witness beam characteristics as a function of the beam propagation distance $s$. (**a**) The average beam energy (blue lines) and the relative energy spread $\sigma_{\gamma_e}/\langle\gamma_e\rangle$ (black lines). $\langle\gamma_e\rangle$ is average Lorentz gamma factor that represents the beam energy. (**b**) The RMS transverse size $\sigma_r$ (blue lines) and the normalized projected emittance $\epsilon_n$ (black lines). Results of both the low-charge (120 pC, denoted by dash-dotted lines) and high-charge (400 pC, denoted by solid lines) cases are shown.

The transverse wakefield $W_x$ is shown in Figure 2b, which is focusing for the witness electrons. However, as shown in Figures 1 and 2b, the transverse wakefield varies continuously in $\xi$-direction before the bubble area. The plasma bubble with linear focusing force only covers the rear part of the witness beam. This happens since the electron bunch takes time to expel the plasma electrons from its propagation axis, with the exact dynamics subject to the beam charge distribution. Therefore, for a moving beam, the plasma bubble trails behind its low density head. At the head of the witness bunch where the plasma electron blow-out is incomplete, the witness electrons are exposed to the relatively weak plasma wakefield with non-linearity in both directions.

The initial witness radius is matched to the strong focusing fields in the bubble, which causes the head of the witness beam, where the fields are weaker, to diverge. As the focusing field varies in $\xi$, the radius of each beam slice oscillates at different local betatron frequencies. The phase difference of oscillations between different slices of the beam then leads to an increase in the projected (whole beam) radius and normalised emittance, as shown in Figure 3. Nonlinear focusing fields may also contribute to the emittance growth.

The emittance and radius of the witness beam continue to grow over the first few tens of centimetres until the full phase-mixing. After reaching their maximum values, the normalised emittance remains essentially constant over the remaining acceleration length, while the RMS radius decreases due to adiabatic damping [21], following the scaling law $\sigma_r \propto \gamma_e^{-1/4}$. Here $\gamma_e$ is the Lorentz gamma factor of the witness beam.

As a solution for the incomplete blowout of the plasma electrons at the head of the witness, one can increase the charge of the witness beam to a higher value [9], e.g., 400 pC. Here we retain the initial transverse bunch size $\sigma_{r0} = \sigma_{r,ic}$ but increase the bunch length $\sigma_{ze}$ from 60 µm to 100 µm due to the requirement of flattening the accelerating field $E_z$. It is shown in Figure 3 that the witness beam with 400-pC charge has a lower emittance growth and smaller beam size after acceleration. However, this benefit is compromised by a lower energy gain and increased relative energy spread due to the over loading on the driver's wakefield.

## 4. Influences of Non–Gaussian Transverse Distributions

### 4.1. Axisymmetric Non–Gaussian Transverse Distribution

Realistic non-Gaussian beam distributions can be simply classified into two categories: the axisymmetric and non-axisymmetric distributions [11]. Here, we first look at the witness beam acceleration with axisymmetric non-Gaussian transverse distributions.

Beams with axisymmetric transverse profiles can be fitted by the super Gaussian (SG) function (or generalized Gaussian) in the form of $f(x) \propto e^{-|x|^p}$, where $p$ is the form parameter [11,13]. The one-dimensional (1D) projection of five different transverse beam density profiles in SG distributions are shown in Figure 4. They have the same initial RMS radius $\sigma_{r0} = \sigma_{r,ic}$ as the baseline Gaussian case ($p = 2$) at the injection point.

Using the same simulation configuration as the Gaussian case, we study how these axisymmetric SG density distributions affect the witness beam acceleration in the quasi-linear plasma wakefield. In Figure 5, the dependence of the normalized transverse beam emittance at the end of the acceleration ($s = 10$ m) on the form parameter $p$ is shown. It is found that the evolution of the normalized emittance of different cases generally follow that of the Gaussian case, i.e., the trend shown in Figure 3. The final value of emittance depends weakly on the distribution shape. Nonetheless, the transverse emittance of all non-Gaussian cases is shown to be slightly lower than that of Gaussian beam ($p = 2$).

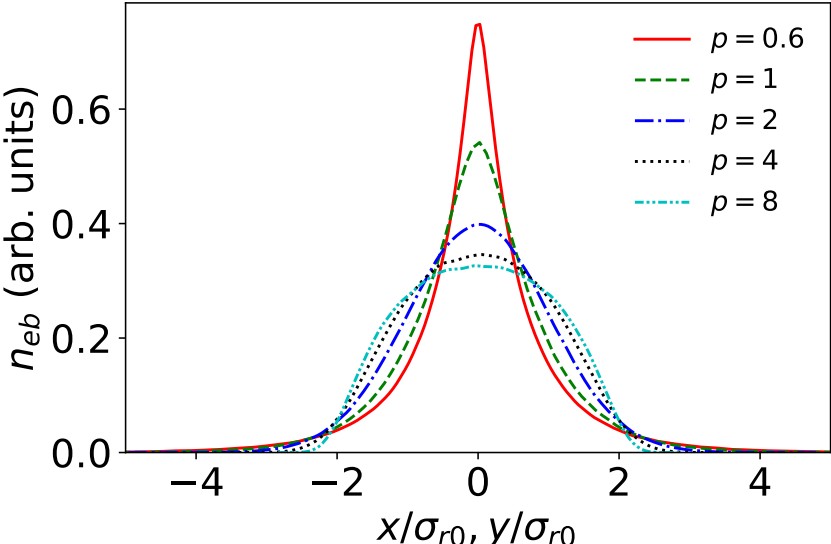

**Figure 4.** 1D transverse beam density distributions in generalized Gaussian function. $p$ is the form parameter. All distributions have the same standard deviation $\sigma_{r0}$.

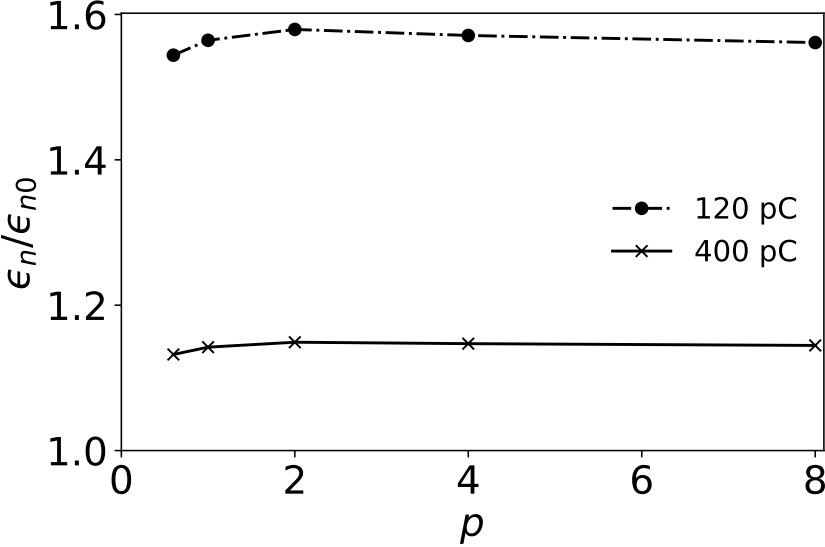

**Figure 5.** Dependence of the normalized transverse beam emittance $\epsilon_n$ at $s = 10$ m on the form parameter $p$ of the initial SG distributions for both charge groups (120 pC and 400 pC).

In order to characterise these non-Gaussian distributions, higher-order statistical parameters such as the kurtosis can be used [22,23]. As shown in Figure 4, for cases $p < 2$, these distributions have longer tails but highly peaked core densities, which correspond to a larger kurtosis. For more rectangular distributions with $p > 2$, they have a smaller kurtosis. The kurtosis of a beam's transverse spatial profile is a good indicator of the visually-observable halo, i.e., a low-density ring surrounding the higher-density core. However, the spatial profile kurtosis will oscillate when the transverse phase-space ellipse of the beam rotates [23]. In the transverse focusing field, the transverse motions of beam particles are governed by $dp_x/dt = -K^2 x$ and $dx/dt = p_x/\gamma m_0$, where $K$ is the focusing strength. The collective motion of particles allows the spatial profile and the momentum profile of the beam to be coupled and mixed during the beam propagation, which then leads to oscillations of the beam's spatial profile kurtosis as well as the momentum kurtosis.

The beam halo in the 2D transverse phase-space $(x, x')$ can be described by the halo parameter $H$ as defined by Equation (A1). For those SG spatial profiles with form parameters of $p = 0.6, 1, 2, 4$ and 8, their halo parameters are calculated as 2.05, 1.68, 1.0, 0.75 and

0.65, respectively, if the corresponding momentum profiles are Gaussian. According to these values and the results shown below, a halo parameter larger than 1 means that the beam has highly-peaked non-Gaussian transverse profiles in at least one dimension of the transverse phase-space, while for $H < 1$, the beam has a flat and low-density profile in at least one dimension.

The evolution trend of the beam halo parameter $H_x$ of the above cases during the beam propagation is shown in Figure 6a,b. The $y$-direction halo parameter $H_y$ is essentially equal to $H_x$ in the 2D cylindrical geometry. Similar to the evolution of the normalized transverse emittance shown in Figure 3b, the halo parameters of different initial beam profiles see a rapid absolute increase at the early stage and then evolve slowly with the beam propagation. However, unlike the emittance, the final beam halo parameters strongly depend on the initial distributions. Beams with a high initial halo parameter, i.e., sharply peaked with a low $p$-factor, results in a higher final halo parameter. These effects are true for both charge groups. The difference due to different witness beam charge is shown in the early-stage evolution of the halo parameter. The low-charge beams see a much larger initial growth of the halo parameters than high-charge beams. This is related to the weak plasma focusing force outside the plasma bubble. It is also due to the same reason, the halo parameter of the low-charge cases drops after reaching the peak as halo electrons with large transverse momentum leave the simulation window and are lost.

To better understand the evolution of the beam halo parameter, the slice distributions of the halo parameters are analysed, as shown in Figure 6c. Here, results of the high-charge cases are shown for example. As shown in Figure 1, the plasma bubble does not cover the whole witness beam. The witness electrons inside and outside the bubble experience different strengths of the plasma focusing force. This then generates different beam slice halo parameter evolution. For particles at the rear of the witness beam, their halo parameters are generally consistent along $\xi$, while for beam electrons at the head, the slice halo parameter varies with the longitudinal location of each transverse slice. Since the halo parameter is an invariant under the linear transverse focusing force according to Equation (A2), we expect that the slice halo parameters are generally "preserved" in the bubble's field. However, it is shown in Figure 6c that the halo parameters of witness electrons inside the bubble still see a small increase after the acceleration, which is due to the minor non-linearity in the loaded transverse wakefield.

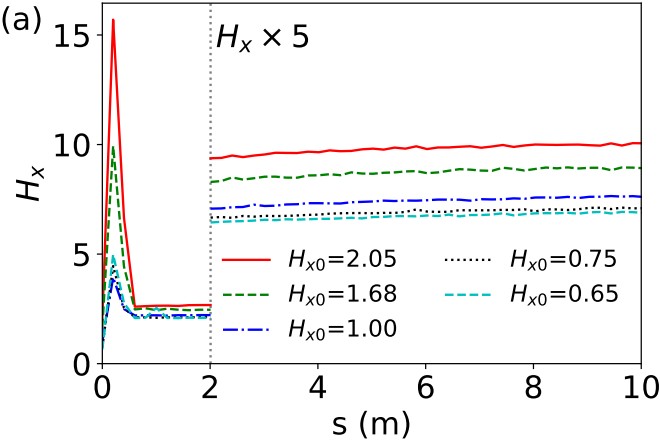
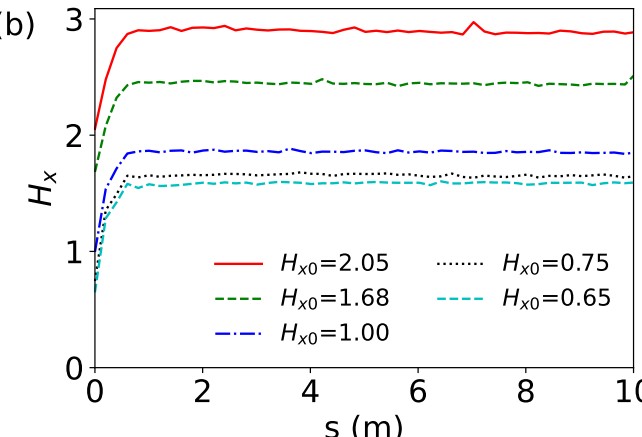

**Figure 6.** *Cont.*

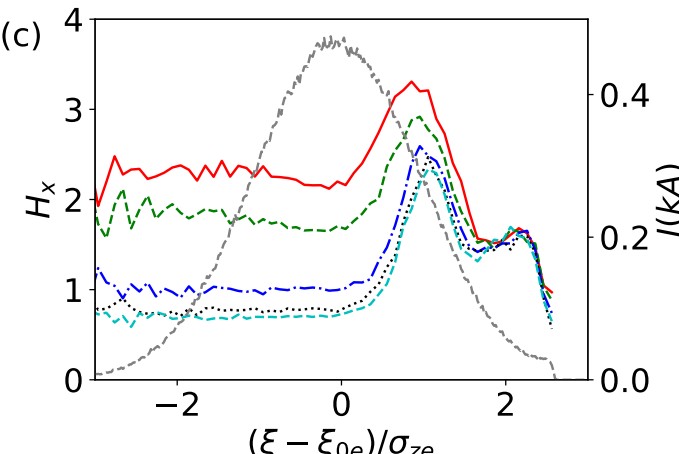

**Figure 6.** Halo parameters of witness beams with SG spatial profiles. Evolution of the transverse profile halo parameter $H_x$ for the (**a**) low-charge (120 pC) cases and (**b**) high-charge (400 pC) cases are presented. (**c**) The slice distribution of the high-charge beams' halo parameter $H_x$ at the end of acceleration ($s = 10$ m). Slice bin size is 5dz and $dz = 0.01k_p^{-1}$ is the longitudinal grid resolution. The longitudinal coordinates of beam slices are relative to the initial centroid position $\xi_{0e}$ and normalized by the initial witness bunch length $\sigma_{ze}$. The absolute current profile $I$ is shown by the grey dashed line.

For head slices around $\xi_{0e} + 2\sigma_{ze}$, i.e., near the leading edge of the witness beam, we can see that local slice halo parameters for beams with different initial spatial profiles converge after the acceleration. This suggests that after reaching full phase-mixing in the non-linear transverse wakefield, these beams have the similar transverse distributions at the head. It should be noted that at the front tip of the high-charge beam, these witness electrons are decelerated and slip backward since the bunch is too long to be accommodated by the accelerating phase. This causes the head erosion, then leads to the reduction of the local slice halo parameter.

For beam slices at around $\xi_{0e} + \sigma_{ze}$, one can see a larger increase of the slice halo parameter in Figure 6c. In this region, the plasma bubble starts to form but the local bubble radius is smaller than that of the witness beam. Although a majority of those witness electrons stay inside the bubble and are focused by the strong bubble-regime wakefield, there is still a small portion of the witness charge falling outside the bubble. The dense sheath of plasma electrons at the edge of the bubble shields the plasma ion charge, leading to the reduction of the plasma focusing strength outside. This huge disparity in the plasma focusing strength leads to a larger increase of the local halo parameter than at other positions along the witness bunch. It is also the main contribution for the whole beam halo parameter increase in the first metre. For the 120-pC beams, we observe a similar evolution of the slice halo parameters. However, as the plasma bubble takes longer to form than in the high-charge case, the initial halo parameter growth is much more significant.

According to the results shown above, whole-beam statistics such as the emittance are easily dominated by particles in the halo. However, the emittance can always be reduced by removing these particles with large betatron oscillation amplitudes. It is therefore more convenient to consider only the particles in the core. A spatial range of the core can be chosen as the matched beam radius $\sigma_{r,ic}$ in the plasma ion column. Similarly, the momentum range for the core can be chosen as the matched transverse momentum spread, $\sigma_{p_r,ic} = \epsilon_{n0}/\sigma_{r,ic}$. Considering these two factors, we define the core range of the beam in the 4D transverse space $(x, p_x, y, p_y)$ as where witness electrons satisfy the condition of

$$\frac{x^2 + y^2}{\sigma_{r,ic}^2} + \frac{p_x^2 + p_y^2}{\sigma_{p_r,ic}^2} < 4. \tag{2}$$

This condition allows the evolution of the charge fraction within this core range to be measured compared with an initially fully-matched beam. For the ideal case of a fully matched beam, the charge in the core should remain constant over acceleration, in a similar way as the emittance.

Figure 7 shows the charge fraction $Q_{core}/Q_0$ within the core range for the considered initial spatial profiles. In Figure 7a, $Q_{core}/Q_0$ first sees a rapid decrease due to the initial expansion of the beam radius. Then after the reaching full phase-mixing of the transverse electron oscillations, $Q_{core}/Q_0$ remains almost constant except for a minor increase.

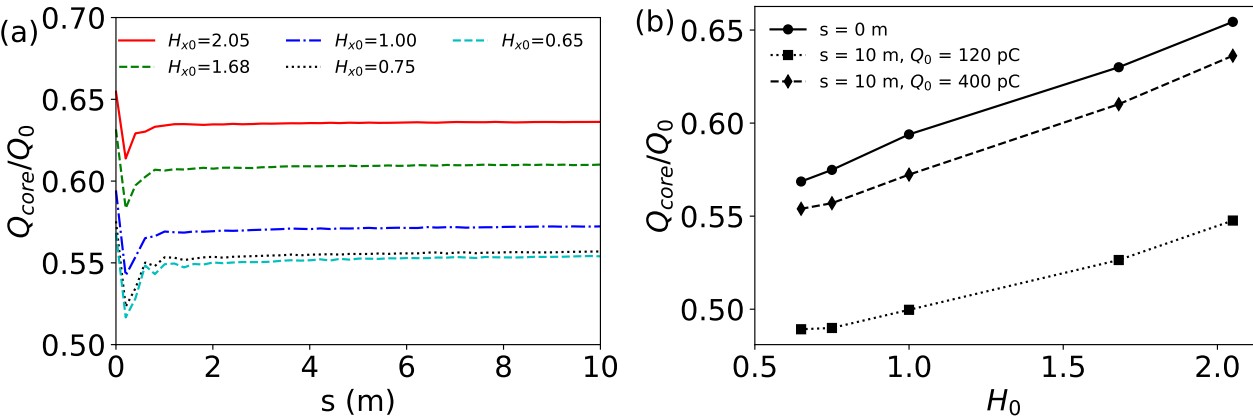

**Figure 7.** (**a**) Evolution of the charge fraction $Q_{core}/Q_0$ within the core range of 400-pC witness beams. Different SG spatial profiles are denoted by their initial halo parameters $H_0$. (**b**) Results for both the 120-pC and 400-pC beams. The initial value of $Q_{core}/Q_0$ (denoted by the black solid line with dots) is the same for both charge groups.

As can be seen from Figure 7b, the fraction of charge in the beam core after acceleration increases monotonically with the initial beam halo parameter $H_0$ for both charge groups. As the increase of charge in the core contributes to the increase of the core brightness, these results also suggest that the beam with a large transverse profile halo parameter at injection can maintain a higher brightness after acceleration. Since the 400-pC witness beams generate stronger transverse focusing field than 120-pC beams, the final value of $Q_{core}/Q_0$ is also slightly higher for 400-pC witness beams. It should be noted that the initial value of $Q_{core}/Q_0$ is independent of the beam charge since they have the same distributions.

Above we have discussed the scenario where the beam's transverse spatial profiles follow the super Gaussian (SG) distribution, while their momentum profiles are still Gaussian. Figure 8 presents the opposite situation where the beam's initial transverse momentum profiles exhibit different SG distributions, while the spatial profiles are Gaussian. For the two equivalent cases with $H_0 = 1$, i.e., with Gaussian profiles in all dimensions, we get the same results in all aspects as expected. However, the impact of the SG spatial profiles and momentum profiles are not entirely the same.

In Figure 8a, we can see that the beam emittance has a stronger dependency on the initial momentum profiles than on the initial spatial profiles. For the initial low-kurtosis transverse momentum profiles ($H_0 < 1$), these beams exhibit a larger emittance than the SG spatial profile beams with $H_0 < 1$ as well as all cases with $H_0 > 1$. For the initial beams with highly-peaked momentum profiles ($H_0 > 1$), they show a slightly lower emittance than their spatial profile counterparts.

Nonetheless, Figure 8b shows that the SG momentum profiles can generate the same effect in the core-range charge fraction as the SG spatial profiles. The initial value of $Q_{core}/Q_0$ converges for both cases since they are equivalent according to Equation (2). After acceleration, the beam with a larger halo parameter $H_0$ also shows a higher $Q_{core}/Q_0$, which is true for both cases. As discussed above, this is due to the coupling of the transverse particle position and momentum through the transverse particle motion in the focusing

plasma wakefield. As a result, the particle distribution can be transferred between the two dimensions.

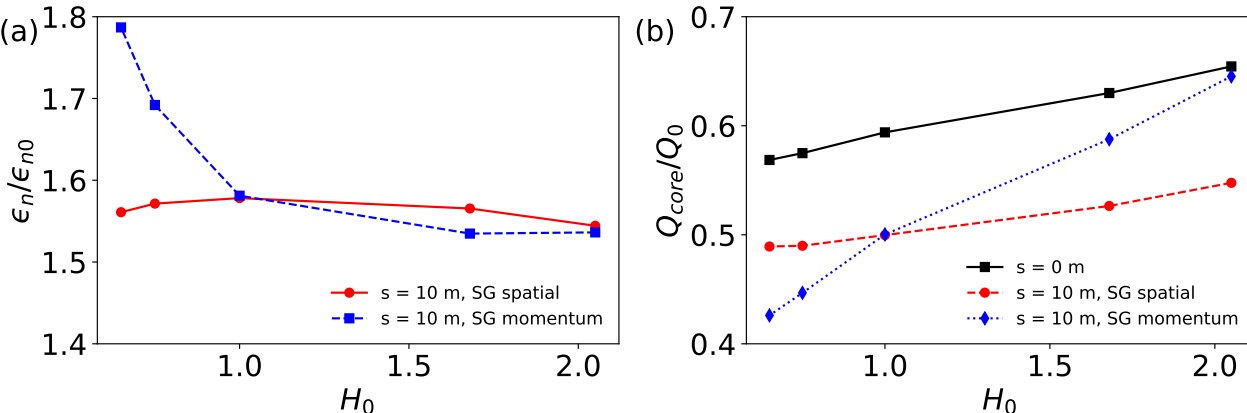

**Figure 8.** Comparison of the influences of initial SG spatial profiles and the SG momentum profile. Results of the low-charge (120 pC) witness beams are shown. (**a**) The normalized transverse emittance $\epsilon_n$. (**b**) The charge-in-core ratio $Q_{core}/Q_0$.

However, when we compare the final value of $Q_{core}/Q_0$ for beams with the same initial halo parameter $H_0$, the SG spatial profiles and SG momentum profiles have different impacts. In Figure 8b we can see that for beams with $H_0 > 1$, the highly-peaked transverse momentum profiles at injection result in a higher charge in the core than the corresponding spatial profile case. Conversely, for beams with a low-kurtosis momentum profile ($H_0 < 1$), they have a lower $Q_{core}/Q_0$ than the beam with a SG spatial profile. These results are related to the initial evolution of the beam where the momentum profiles show a more significant impact.

*4.2. Non-Axisymmetric, Non-Gaussian Transverse Distribution*

For non-axisymmetric distributions, a simple correction to the standard Gaussian is the skew-normal (SN) function [24]: $f(x) = 2\phi(x)\Phi(\alpha x)$, where $\alpha$ is its form parameter, $\phi(x) = (2\pi)^{-1/2}e^{-x^2/2}$ is the standard Gaussian with a cumulative distribution function of $\Phi(x) = 0.5\left[1 + \text{erf}\left(x/\sqrt{2}\right)\right]$ and $\text{erf}(x) = 2\pi^{-1/2}\int_0^x e^{-t^2}\,dt$ is the error function. Here we only consider the SN distribution in the $x$-direction as shown in Figure 9a, while the $y$-direction beam profile is still in the standard Gaussian. It should be noted that the mean position in the $x$-direction is still zero, i.e., on the axis.

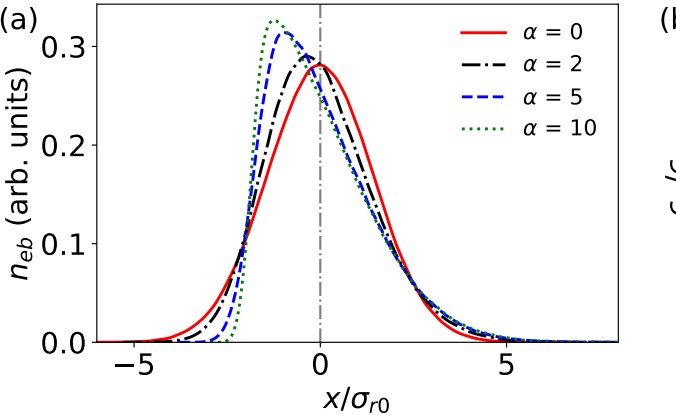
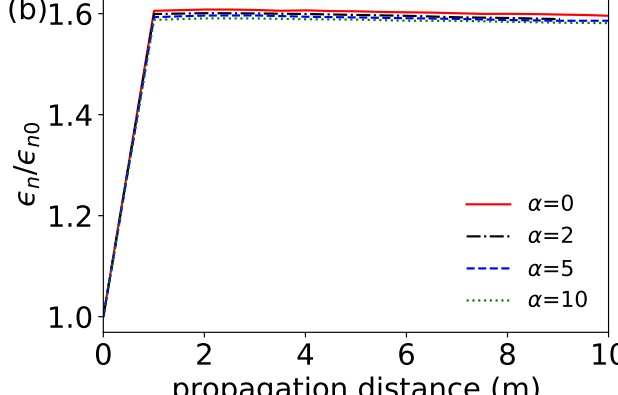

**Figure 9.** *Cont.*

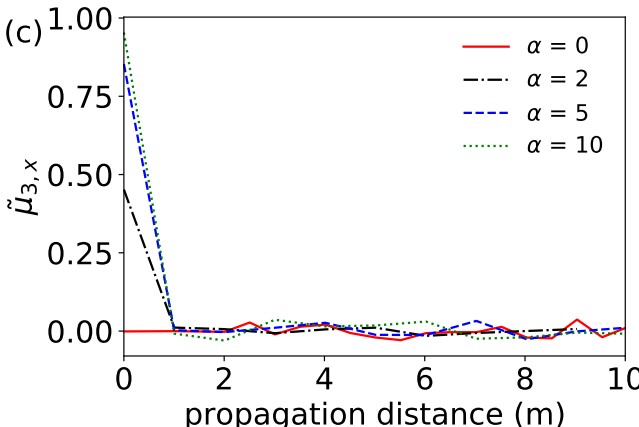

**Figure 9.** Influence of the skew-normal (SN) transverse beam density profile. (**a**) The 1D density profile in the *x*-direction at injection. $\alpha$ is the form parameter of the SN distributions. $\tilde{\mu}_{3,0}$ and $h_0$ is the skewness and kurtosis. (**b**) Evolution of the normalized transverse emittance $\epsilon_n$. (**c**) Evolution of the skewness of the *x*-direction beam density profile. Data points are sampled per half-metre.

To measure the degree of asymmetry of a distribution, the skewness $\tilde{\mu}_3 = \langle x^3 \rangle / \langle x^2 \rangle^{3/2}$ can be used. As illustrated by Figure 9c, the initial skewness $\tilde{\mu}_3$ increases with the form parameter $\alpha$ of the SN distributions. In the limit when $\alpha \to \infty$, the skewness reaches the maximum of 1 and the SN distribution becomes triangular. The transverse asymmetry can arise due to the transverse wakefields induced by an off-axis beam in the RF cavities, higher-order dispersion such as the quadratic T166 term, as well as potential well distortion [11].

Since this problem is non-axisymmetric, for these simulations we use the fully 3D code QV3D [25], built on the VLPL platform [26]. Similar to the case of super Gaussian distributions, Figure 9b shows that SN distributions have only a very small impact on the beam energy gain and the normalized RMS beam emittance $\epsilon_n$. The beam emittance growth after saturation is only slightly smaller with the increase of the SN distribution form parameter $\alpha$. It is also interesting to see that in Figure 9c the *x*-direction skewness $\tilde{\mu}_{3,x}$ soon damps and becomes almost negligible after a propagation distance of 1 m.

The above results suggest that a degree of asymmetry of a beam transverse distribution is allowed for the witness beam acceleration in the studied case, and it can be automatically corrected by the plasma response. This is consistent with previous studies for the beam injection with minor transverse offsets [7]. It shows that the emittance of the witness electrons inside the bubble is also not strongly affected. This is due to the decoherence of the betatron oscillations along the bunch as described in Section 3, which helps to suppress the hosing instability [27,28].

## 5. Conclusions

In this study, we find that the normalised transverse emittance of the witness beam does not depend strongly on the initial shape of the beam's transverse distribution. However, the halo parameter of the transverse beam profile shows a clear dependence on the initial shape of the beam at the injection point. This is mainly due to the conservation of the halo parameter in the linear focusing region, especially inside the plasma bubble. An increase of the projected halo parameter arises due to the non-linear focusing of the beam head outside the bubble area.

We also show that a beam with an axisymmetric non-Gaussian transverse momentum profile can have a stronger influence on the beam quality, impacting both the emittance and the charge in the beam core after acceleration.

In addition to the axisymmetric transverse beam distributions, electron acceleration with a non-axisymmetric transverse profile is also studied. It is shown that an initially non-axisymmetric beam with transverse mean position on axis becomes symmetric due to

the interaction with the plasma wakefield, and so it does not cause a detrimental effect for the beam acceleration.

**Author Contributions:** Conceptualization, L.L. and J.P.F.; methodology, L.L. and J.P.F.; software, A.P. and J.P.F.; validation, L.L. and J.P.F.; formal analysis, L.L.; investigation, L.L.; resources, G.X. and J.P.F.; data curation, L.L.; writing—original draft preparation, L.L.; writing—review and editing, L.L., G.X. and J.P.F.; visualization, L.L.; supervision, G.X. and J.P.F.; project administration, G.X.; funding acquisition, G.X. All authors have read and agreed to the published version of the manuscript.

**Funding:** This research was funded by Cockcroft Institute Core Grant and the STFC AWAKE Run 2 grant ST/T001917/1.

**Institutional Review Board Statement:** Not applicable.

**Informed Consent Statement:** Not applicable.

**Data Availability Statement:** Not applicable.

**Acknowledgments:** The author would like to thank Konstantin Lotov's group for their kind help on the use of LCODE. Computing resources are provided by the SCARF HPC of STFC and the CERN batch services.

**Conflicts of Interest:** The authors declare no conflict of interest.

## Appendix A. Halo Parameter

The halo parameter $H$ generalizes the spatial-profile kurtosis $h$ to the 2D phase-space $(x, x')$ using the kinematic invariants of the particle distribution, which is given as [23]:

$$
\begin{aligned}
H &= \frac{\sqrt{3I_4}}{2I_2} - 2, \\
I_2 &= \left\langle x^2 \right\rangle \left\langle x'^2 \right\rangle - \left\langle xx' \right\rangle^2, \\
I_4 &= \left\langle x^4 \right\rangle \left\langle x'^4 \right\rangle + 3\left\langle x^2 x'^2 \right\rangle^2 - 4\left\langle xx'^3 \right\rangle \left\langle x^3 x' \right\rangle,
\end{aligned}
\tag{A1}
$$

where $\langle x^m x'^n \rangle = \frac{1}{N} \sum_{i=1}^{N} (x_i - \bar{x})^m (x'_i - \bar{x}')^n$, $\bar{x}$ and $\bar{x}'$ are the mean values of $x$ and $x'$, respectively. One can see that $I_2$ factor is exactly the product of the trace-space beam emittance.

In the linear focusing system with a uniform focusing strength of $K$, where $x'' = -K^2 x$, the halo parameter $H$ is an invariant, since

$$
\frac{\mathrm{d}H}{\mathrm{d}z} = \frac{3}{4I_2\sqrt{I_4}} \frac{\mathrm{d}I_4}{\mathrm{d}z} - \frac{\sqrt{3I_4}}{2I_2^2} \frac{\mathrm{d}I_2}{\mathrm{d}z} = 0,
\tag{A2}
$$

where the kinematic invariants $I_2$ and $I_4$ are positive figures with derivations of

$$
\begin{aligned}
\frac{\mathrm{d}I_2}{\mathrm{d}z} = \frac{\mathrm{d}\epsilon_x^2}{\mathrm{d}z} =& \frac{\mathrm{d}}{\mathrm{d}z}\left( \left\langle x^2 \right\rangle \left\langle x'^2 \right\rangle - \left\langle xx' \right\rangle^2 \right) \\
=& 2\langle xx' \rangle \left\langle x'^2 \right\rangle + 2\left\langle x^2 \right\rangle \left\langle x'x'' \right\rangle \\
& - 2\left\langle xx' \right\rangle \left\langle x'^2 \right\rangle - 2\left\langle xx' \right\rangle \left\langle xx'' \right\rangle \\
=& 0,
\end{aligned}
\tag{A3}
$$

and

$$
\begin{aligned}
\frac{\mathrm{d}I_4}{\mathrm{d}z} =\ & 4\left\langle x^3 x'\right\rangle\left\langle x'^4\right\rangle + 4\left\langle x^4\right\rangle\left\langle x'^3 x''\right\rangle \\
& + 12\left\langle x^2 x'^2\right\rangle\left\langle xx'^3\right\rangle + 12\left\langle x^2 x'^2\right\rangle\left\langle x^2 x' x''\right\rangle \\
& - 4\left\langle x'^4\right\rangle\left\langle x^3 x'\right\rangle - 4\left\langle xx'^3\right\rangle\left\langle x^3 x''\right\rangle \\
& - 12\left\langle xx'^2 x''\right\rangle\left\langle x^3 x'\right\rangle - 12\left\langle xx'^3\right\rangle\left\langle x^2 x'^2\right\rangle \\
=\ & 0.
\end{aligned}
\tag{A4}
$$

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
