# Peer review of "Acceleration of an Electron Bunch with a Non–Gaussian Transverse Profile in Proton-Driven Plasma Wakefield"

_applsci, doi:10.3390/app122110919_

Round 1

Reviewer 1 Report

The manuscript Acceleration of an Electron Bunch with a Non-Gaussian Transverse Profile in Proton-driven PlasmaWakefield has reported a systematic study on the effect of non-Gaussian transverse trailing beam profile for the proton-driven plasma wakefield accelerator (PWFA). It has used a few parameters to describe the non-Gaussian profiles including different kurtosis, halo parameters, and skewness. Some phenomenological conclusions have been made by using particle-in-cell simulations, and two charge groups have been considered. Although it is not perfect, this study is worthy to be published because there is too few realistic transverse beam profile study in PWFA as far as I know.

I have one question that the authors may try to answer. In PWFA, the hosing instability (or beam breakup instability) is a major limitation for high-quality acceleration. Even for very symmetric initial condition, a small seed of non-symmetry may grow exponentially and largely degrade the beam quality or even beak up the beam. The axisymmetric code may not show any hosing, but the full-3D code may be able to show it. From Fig. 9, it seams the hosing is not a problem at all even for a very non-symmetric initial condition. The authors may show more details and explain why.

Author Response

Thanks to the reviewer for this very good question. In fact, we have observed hosing-like instability in our simulation using the three-dimensional QV3D code. This effect happens when the loaded accelerating gradient is almost constant along the bulk of the witness beam. In this case, witness electrons oscillate in the same betatron frequency, which contributes to the coherent condition for the growth of the hosing instability. With this beam loading condition, we can observe the hosing-like effect even for the axisymmetric non-Gaussian beam. In order to suppress the hosing-like instability, we changed the beam loading position to produce a tilted accelerating gradient profile to break the coherent betatron oscillation, as shown in Fig. 1. This mechanism is like the BNS damping in conventional accelerators [1]. As witness electrons at different locations initially oscillate with different betatron frequencies, the hosing-like effect do not happen again in the simulation. Additionally, since the witness beam soon catches up and surpasses the speed of the proton driver, the witness beam is shifting forward with respect to the driver during the acceleration. The initial difference of accelerating gradient along the beam is then automatically cancelled by the forward phase-shift of the witness beam. In this case, we obtain a lower energy spread of the witness beam than the case with initially uniform accelerating gradient.

It should be noted that the above results may suggest the existence of the hosing instability in our simulation. However, as hosing effect in numerical simulation can also arise from the stochastic fluctuation of the EM fields, we therefore hold conservative point of view about the simulation results with hosing effect and decide not to present these results in this paper. Further study on this point will be carried out in the next step.

[1] V. E. Balakin, A. V. Novokhatsky, and V. P. Smirnov. VLEPP: Transverse beam dynamics. Conf. Proc. C, 830811:119–120, 1983.

Reviewer 2 Report

The manuscript “Acceleration of an Electron Bunch with a Non-Gaussian Transverse Profile in Proton-driven Plasma Wakefield”, by Lindo Liang et al., reports results of numerical simulations applied to a model of the AWAKE Run 2 setup. More specifically, the foreseen Run 2 witness bunch parameter are employed to simulate the electrons acceleration in plasma looking for final performances in terms of emittance, energy and energy spread. Upon noticing the formation of a witness driven plasma bubble due to beam loading, a second, “high charge” baseline setup is also simulated. Moreover, a detailed study of Super Gaussian witness profiles, both in partial and transverse momenta distributions and characterized by the “halo parameter”, is reported, comparing again final emittances and the amount of “core” charge, defined as the charge within a radius equal to the matched one.

The manuscript is well written and easily understood. The simulations setup is adequate for the experimental setup at hand and the findings are clearly explained and supported by numerical results.

Interest is mainly to people involved in the AWAKE experiment, although most features are in common with more general experimental setups.

Overall, the novelty level is low, since most findings are well known textbook results: a matched bunch with cylindrical symmetry in a plasma channel is (almost) always an emittance dominated beam, so that its transverse equilibrium distribution is a Gaussian both in positions and momenta. If this equilibrium is somehow “slightly” perturbed at injection, the bunch will approach a new equilibrium with a larger emittance; otherwise, it will eventually breakup as charges escape from the accelerating/focusing bucket [see, for example M. Reiser, Theory and Design of Charged Particle Beams, Wiley]. Even the formation of a beam loading generated bubble improving linearity of the transverse force was already reported in [Phys. Rev. Acc. Beams 23, 071301 (2020)] (although in a slightly different context).

That being said, I do not see any impairment in granting publication, provided Authors cope with the following minor points:

a)  the introduction of the higher charge witness is meant to improve the beam loading generated bubble by increasing the density ratio between bunch and unperturbed plasma. However, when this ratio is much larger than one, a more significant parameter for determining the generated bubble regime, is the normalized charge [Phys. Rev. STAB 7, 061302 (2004)]. Authors please include the evaluation of this parameter.

b) The aforementioned papers need to be included into the bibliography and properly cited.

c) Authors, please consider if the larger energy spread for the higher charge witness is only due to an excess of beam loading or if the larger bunch length also contributes to this result.

d) All figures are rather small and need to be enlarged.

e) Authors, please consider expanding the explanation of how and why a Super Gaussian momenta distribution affects emittance and core charge.

Author Response

Thanks reviewer for many great comments about our work. We have tried to answer your questions in a separate pdf file, as attached.
